# Design and Fabrication of a MEMS Bandpass Filter with Different Center Frequency of 8.5–12 GHz

**DOI:** 10.3390/mi14020280

**Published:** 2023-01-21

**Authors:** Yi-Fei Zhang, Min Cui, Dong-Ping Wu

**Affiliations:** 1State Key Laboratory of ASIC and System, Fudan University, Shanghai 200433, China; 2School of Microelectronics, Fudan University, Shanghai 200433, China; 3School of Instrument and Electronics, North University of China, Taiyuan 030051, China

**Keywords:** RF MEMS switch, filter

## Abstract

The design simulation and fabrication results of a bandpass filter based on micro-electro-mechanical system (MEMS) switches are presented in this paper. The MEMS filter element consists of a MEMS capacitance switch and two resonant rings that are fixed onto coplanar waveguide lines through anchor points. The micromachine characteristics of the filter could be optimized to change the center frequency from 8.5 to 12 GHz by improving the geometrical parameters; other electrical parameters of the filter, such as stopband rejection, insertion loss, and return loss at each center frequency, were simulated and calculated. In order to evaluate the MEMS filter design methodology, a filter working at 10.5 GHz fabricated with an aluminum top electrode was used, and it displayed a low insertion loss of 1.12 dB and a high stopband rejection of 28.3 dB. Compared with the simulation results, these proposed filter showed better electrical performance. Our results demonstrated that the filter with the integrated RF MEMS switch not only provides the benefit of reduced size compared with a traditional filter, but also improves stopband rejection, insertion loss, and return loss.

## 1. Introduction

In recent years, filters have become a new research hotspot, and some filters based on new materials and new theories have emerged, such as filters based on graphene materials [1], filters based on silicon nitride [2,3], filters based on photon quantum theory [4,5], the mathematical model of filters [6], and filters integrated with a micro-electro-mechanical system (MEMS) switch [7,8,9,10]. The radio frequency of a micro-electro-mechanical system (RF MEMS) tunable filter, which is a type of electronic component that controls signals by RF MEMS switches, and offers higher integration, lower power dissipation, smaller size, better linearity, and better performance in frequency adjustment than conventional filters [11]. Therefore, over the last two decades, RF MEMS filters have had tremendous applications in modern communications systems, such as satellite communication, radar, medical electronics, and software [12,13,14,15,16,17,18,19,20,21]. As an important electronic device in signal transceivers, the tunable filter, without noise or various unwanted signals in each frequency band, reduces inter-channel interference, ensures the normal operation of communication equipment, and achieves high-quality communication [22]. Electronic devices that require RF filters to have low insertion loss and numerous RF MEMS filters, such as radar and microwave test equipment [23], have thus been proposed and fabricated in recent years. The University of Michigan has designed a tunable filter with four adjustable frequencies, but its out-of-band rejection is only 4–5 dB [24]. The University of California, San Diego, has designed a filter with a smaller volume and an out-of-band rejection greater than −30 dB, but its insertion loss is large (greater than −2 dB) [25].

This paper proposes a tunable bandpass filter based on the micro-electro-mechanical system (MEMS) switch. It consists of a single parallel capacitive RF MEMS switch, two resonators, drive circuits, and active control modules. Resonators with different geometrical characteristics can filter electromagnetic waves of different frequencies. The filter of the corresponding resonator is controlled by changing the state of conduction and disconnection of the same RF MEMS switch, thereby achieving adjustable band pass filtering. Because aluminum is a relatively cheap metal material and the full structure is made of aluminum, some structures can be completed in one step, reducing the amount of photolithography. To reduce the difficulty of fabrication and reduce the production cost, we used a new all-aluminum structure to make the filter.

## 2. Design and Simulation of RF MEMS Filter

RF MEMS filters are generally composed of a single coplanar waveguide (CPW) line, an actuation electrode, top electrode, air bridge, and resonant ring. These components allow the RF MEMS switch, which is actuated using a specific amount of static electricity, to control a signal by opening or closing. Here, we propose an all-aluminum tunable RF MEMS filter that consists of a substrate, a CPW line, two actuation electrodes, a top electrode, a resonant ring, two fixed anchors, and two air bridges. As illustrated in Figure 1a., quartz glass with a low relative permittivity is used as the substrate material for the filter to reduce its insertion loss. The CPW lines are fixed on the glass substrate. The resonant ring can produce different resonant frequencies and control the distance between the upper and lower electrodes of the switch, which is the most important structure in the RF MEMS switch and is fixed on the CPW lines.

As shown in Figure 1b, we selected a resonant-ring structure composed of double rings and ground lines to act as the resonant structure to be fixed on the CPW lines, and among them the middle ring is used as an anchor to support the top electrode along with the anchor on the other side. In a double-ended fixed beam RF MEMS switch, the beam is fixed at both ends and is actuated using a voltage that is applied at both ends of the beam. When the maximal voltage (6.48 V) is applied, the displacement reaches its maximum value (1.5 μm). The operational states of the switch are shown in Figure 2.

The RF MEMS switch in the on-state, where no direct voltage is applied between the top electrode and the actuation electrode, is shown in Figure 2a. The coupling capacitance between the top electrode and the signal line is low, and the RF signal from the input is connected with the output. When the switch is actuated, the top electrode is pulled down by electrostatic force to (2/3)${\;g}_{0}$, where${\;g}_{0}$ is the air gap between the top electrode and the actuation electrode. This process achieves direct metal-to-metal contact between the top electrode and the signal line. The RF MEMS switch is turned off, as shown in Figure 2b. In this case, the actuation voltage, *V*_pull-down_, is given by [26]:(1)$$V_{\text{pull-down}}=V\left(\frac{2}{3}g_{0}\right)=\sqrt{\frac{8k}{27\varepsilon_{0}A}g_{0}{}^{3}}.$$here, $\varepsilon_{0}$ is the permittivity of free space and has a value of 8.854 × 10^−12^ F/m, and *A* is the area of the electrodes (on the top). *k* is the effective spring constant of the cantilever beams of the top electrode and is given by:(2)$$k=\frac{1}{4}Ew{\left(\frac{t}{l}\right)}^{3}.$$

The actuation voltage can be defined as the minimum voltage that is required to pull the switch beam of the RF MEMS switch down. One of the design objectives for MEMS devices with actuation capability is to perform using a low actuation voltage, which is dependent on the design of the switch structure. The actuation voltage is dependent on the geometric parameters of the initial gap between the movable structures, the actuation area, and the electrode. The actuation voltage of 6.48 V is obtained in the straight plate type top electrode switch case, as shown in Figure 3.

### 2.1. Theory

In order to meet the actual application requirements, the filter design needs to meet practical parameter specifications, such as cutoff frequency, center frequency, insertion loss, and stopband rejection. Here is a brief introduction to these parameter indicators: [27]

Center frequency $\left(f_{0}\right)$ refers to the intermediate frequency of the filter passband. Generally, for bandpass and bandstop filters, the specific calculation formula is:(3)$$f_{0}=\sqrt{f_{1}\times f_{2}}.$$here, $\left(f_{1}\right)$ and $\left(f_{2}\right)$ represent the frequencies corresponding to 3 dB points on both sides of the passband.

Insertion loss: When the signal passes through the filtered two-port network, it produces a certain energy loss, which can be expressed as the ratio of the output power of the filter to the input power. The specific calculation formula is:(4)$$IL=-10log\frac{P_{\mathrm{out}}}{P_{\mathrm{in}}}=-10log\left(1-{\left|\Gamma_{\mathrm{in}}\right|}^{2}\right).$$here, $P_{\mathrm{in}}$ is the input power,$\;P_{\mathrm{out}}$ is the output power, and $\left|\Gamma_{\mathrm{in}}\right|$ refers to the reflection coefficient at the input end. In general, the dielectric substrate and the high-temperature superconducting film with low-loss corners can greatly reduce the insertion loss and improve the sensitivity of the filter.

Stopband rejection (SR) is a technical indicator used to indicate the anti-interference ability of a filter. The wider the stopband, the better the anti-interference ability. In general, a step impedance resonator is used, the zero point is generated at the parasitic passband, and different coupling structures are designed to widen the stop band to improve the anti-interference ability of the filter.

The groove structure on the middle signal line can be equivalent to a Pi type network, which is composed of an equivalent series capacitor, C_1_, and two parallel capacitors, C_2_. The wires on both sides of the CPW can be equivalent to a series inductor, L_c_, and a shunt capacitor, C_c_. The annular groove structure etched on the coplanar waveguide ground can be equivalent to the parallel inductance, L_a_, due to the magnetic coupling to the signal line. Ignoring the resistance of the MEMS switch, there is a capacitance, C_a1_, between the MEMS switch beam and the signal line, and the beam itself has an inductance, L_a1_, so it can be equivalent to a circuit form in which two sets of capacitance and inductance are connected in series. The specific equivalent circuit diagram is shown in Figure 4.

As shown in Figure 5, L_c_ and L_b_ can be equivalent to an admittance converter, and a hybrid circuit of C_a1_ and L_a1_ can be equivalent to a resonant circuit. By combining the capacitor inductance, it can be simplified to the equivalent circuit diagram shown in Figure 5. Because the L_a1_ is small, the inductance L_a1_ can be ignored, so the series circuits C_a1_ and L_a1_ can be directly represented by C_a1_, while C_o_ = C_c_ + C_a_ − C_2_, C_p_ = C_a_ + C_b_ + C_c_ − C_2_.

In Figure 4, the admittance inverters can be calculated by the following equation [24]:(5)$$J_{01}=\sqrt{\frac{Y_{0}\omega_{0}C_{a}BW}{g_{0}g_{1}\omega_{1}}}={\left.\frac{1}{\omega L_{p}}\right|}_{\omega =\omega_{0}}\;\;\;\;$$
(6)$$J_{12}=\frac{\omega_{0}C_{a}BW}{\omega_{1}}\sqrt{\frac{1}{g_{0}g_{1}}}={\left.\omega C_{1}\right|}_{\omega =\omega_{0}}\;\;\;$$

### 2.2. Microwave Performance

This section presents the design process utilized for the size-dependent RF properties of the MEMS switch, including insertion loss, isolation, and actuation voltage. The simulation model of the RF MEMS switch is illustrated in Figure 6.

First, a CPW line with the dimensions of W/S/W = 100 μm/340 μm/100 μm was designed for operation in the 8.5–20 GHz range. Second, an impedance value of close to 50 Ω was calculated using the LineCalc tool from ADS software (Advanced Design system 2014, Keysight Technologies, Santa Rosa, CA, USA). Finally, to improve the performance of the broadband RF MEMS switch, the switch dimensions for the “off” and “on” states were designed and optimized using High-Frequency Simulation Software (HFSS) version 13.0 for full-wave analysis. These dimensions included the length and width of the top electrode. In this study, the substrate material of the designed broadband MEMS switch was quartz glass, the dielectric layer was Si_3_N_4_, and the other structures (including CPW, top electrode, actuation electrode, etc.) were all aluminum materials with high stability and less loss. It can be seen in Table 1 that the conductivity of aluminum is superior. Although the conductivity of metals such as gold and copper is slightly higher than that of aluminum, their prices are too expensive, so aluminum was selected as the main metal material in this study.

As a unit structure in the filter, the filtering characteristics of a single RF MEMS switch and a single resonator are particularly critical. Therefore, before designing the multi-frequency adjustable bandpass filter, the performance of the filter cell structure should be designed and studied. The design model of the single-band bandpass filter based on the MEMS switch is shown in Figure 3. By optimizing the width of the filter, the center frequency was changed from 8.5 GHz to 12 GHz.

In this study, the single-frequency bandpass filter was modeled in microwave simulation software Ansys HFSS15.0 (Ansys HFSS15.0, Ansys, Canonsburg, PA, USA), and its geometrical parameters were simulated and optimized. The results are shown in Figure 7, and the filtering performance is shown in Table 2.

## 3. Fabrication and Measurement of the Filter

The fabrication of the filter process involved using a five-layer mask, and was completed by five executions of photolithography. The device was fabricated on a high-resistance silicon substrate, and the complete filter processing flow is described below in eight steps. The image of the sample is shown in Figure 8.

(1) Preparation, cleaning, and quartz glass substrate cleaning: the glass was placed in a NH_3_·H_2_O (ammonia) + H_2_O_2_ mixture (according to a volume ratio of 30%NH_3_·H_2_O: 30%H_2_O_2_: H_2_O = 1:1:5) and boiled for 5 min, then rinsed and dried repeatedly with hot and cold deionized water.

(2) The fabrication of CPW: A 2 μm Al film was sputtered as a CPW. The photoresist AZ5214 was spin-coated on the Al film at a speed of 4000 rad/min. The photolithography plate was used as a mask for exposure and development. After that, O_2_ plasma was used to apply the base film for 5 min. A 30% phosphoric acid water bath was used to heat it to 50 ℃ and corrode it for 12 to 15 min. Then, the water was flushed with acetone to remove the photoresist in it.

(3) The fabrication of the dielectric layer: 400 nm of plasma enhanced chemical vapor deposition (PECVD) Si_3_N_4_ was patterned and etched by RIE (Reactive Ion Etching) as the dielectric layer.

(4) The fabrication of resonant ring: 3 μm Al film was sputtered as a resonant ring. The photoresist AZ4620 was spin-coated on the Al film at a speed of 4000 rad/min. The photolithography plate was used as a mask for exposure and development. After that, O_2_ plasma was applied to the base film for 5 min. Then, a 30% phosphoric acid water bath was used to heat it to 50 °C and corrode it for 18 to 22 min. Finally, the water was flushed with acetone to remove the photoresist on the wafer.

(5) Sacrificial layer preparation: Here, 2.9 μm thick polyimide (PI) was spun-cast, solidified, and patterned to define a sacrificial layer.

(6) Making anchor through holes: AZ4620 photoresist was spin-coated on the pre-cured PI film. The speed of the glue was 4300 rpm, the glue time was 30 s, and the thickness of the photoresist was 7~8 μm. A pre-baked AZ4620 was set to a temperature of 85 °C for 45 min at an exposure time of about 2 min. After the exposed photoresist was removed by using the photoresist developing solution, the PI was further etched, and the anchor through the hole was etched for 3~4 min. PI is insoluble in acetone, so the AZ4620 photoresist was removed with acetone to obtain a patterned PI film. The residual AZ4620 on the PI surface and the residual PI at the bottom of the pit were removed by an oxygen plasma stripper for 5 min.

(7) The fabrication of the top electrode: We sputtered 2 μm Al film as a top electrode. The photoresist AZ4620 was spin-coated on the Al film at a speed of 4000 rad/min. A photolithography plate was used as a mask for exposure and development. After that, O_2_ plasma was applied to the base film for 5 min. Then, a 30% phosphoric acid water bath was used to heat it to 50 °C and corrode it for 12 to 15 min. Finally, the water was flushed with acetone to remove the photoresist in it. 

(8) Sacrifice layer release: Finally, the sacrificial layer was released in the atmosphere of oxygen plasma to complete the whole fabrication process.

The sample was added to the test system, and the transmission status (S11, S21), test frequency, and appropriate IF bandwidth were set. For more accurate measurements, the instrument was fully port calibrated by using a calibration tool (the machine model was T40A-GSG500 from MJC Probe Incorporation (MPI) company, and it was calibrated with an AC5 calibration sheet). According to the filter requirements, the maximum resonance point of the filter was selected. Accordingly, the return loss and the insertion loss at the frequency point could be seen on the screen. Comparing the test results with the simulation results, as shown in Figure 6b, we found that the test results were basically consistent with the simulation results, and they exhibited better stopband rejection than the simulation results in the frequency band above the center frequency. The specific results are compared in Table 3 and Figure 9.

Table 4 provides a comparison of the results between other filters and the filter designed in this study. Compared with the current bandpass filters, the filter based on MEMS switches has the advantages of a low insertion loss, a wide range of switching frequency bands, and a high stopband rejection

## 4. Conclusions

In this study, we designed and simulated an RF MEMS filter for applications within the 8.5–12 GHz range. Eight center frequencies were selected within this frequency to carry out a series of device design simulations. The simulation results showed a filter insertion loss of ≤1.27 dB and a stopband rejection of ≥25 dB. In order to evaluate the MEMS filter design methodology, a filter worked at 10.5 GHz was fabricated with an aluminum top electrode, which had a low insertion loss of 1.12 dB and a high stopband rejection of 28.3 dB. Compared with the simulation results, the proposed filter has better electrical performance, including less isolation and stopband rejection. The proposed filter thus provides lower insertion losses and higher stopband rejection than the traditional filter.

## Figures and Tables

**Figure 1 micromachines-14-00280-f001:**
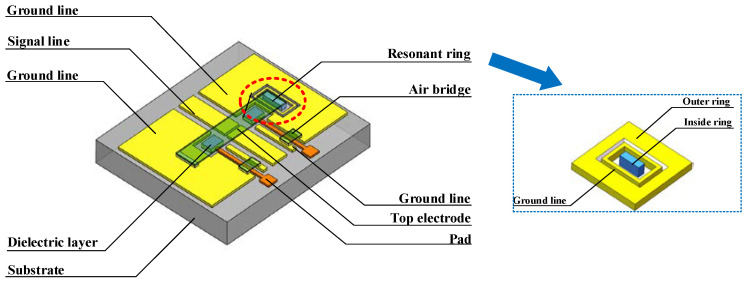
Structure of the RF MEMS filter. (**a**) MEMS filter; (**b**) resonant ring.

**Figure 2 micromachines-14-00280-f002:**
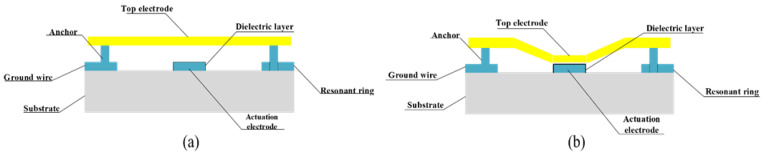
Cross-sectional views of filter operation:(**a**) off state; (**b**) on state.

**Figure 3 micromachines-14-00280-f003:**
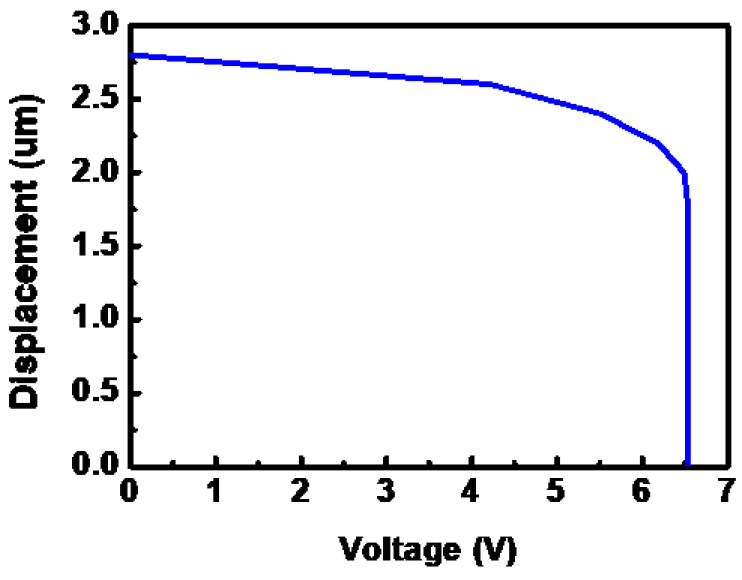
Displacement versus actuation voltage.

**Figure 4 micromachines-14-00280-f004:**
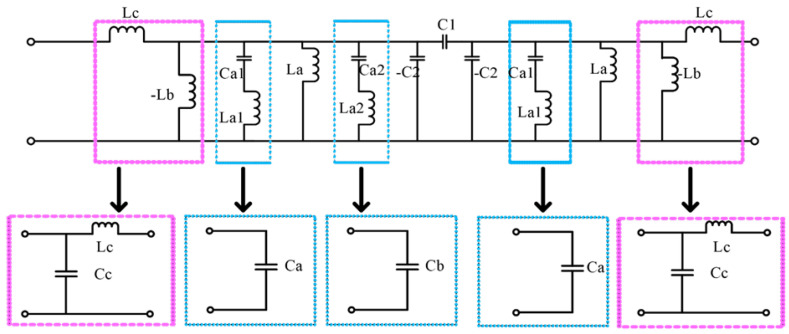
Equivalent circuit of the filter.

**Figure 5 micromachines-14-00280-f005:**
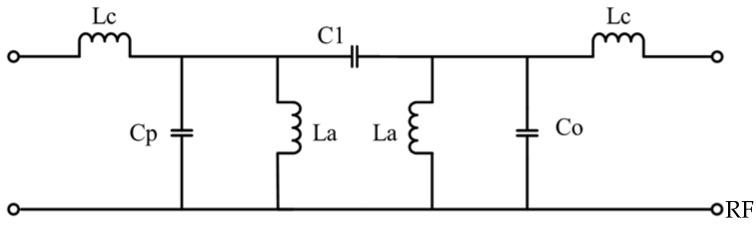
Simplified equivalent circuit.

**Figure 6 micromachines-14-00280-f006:**
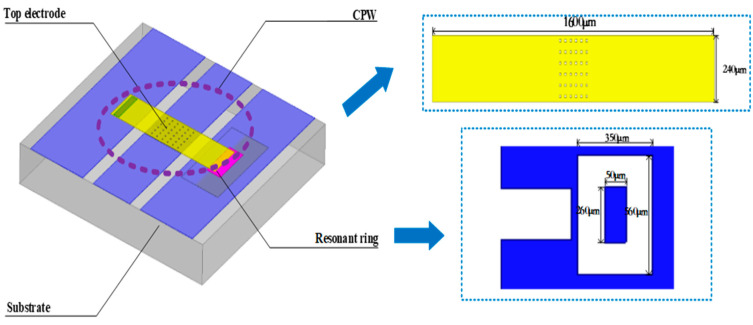
Simulation model of the RF MEMS filter.

**Figure 7 micromachines-14-00280-f007:**
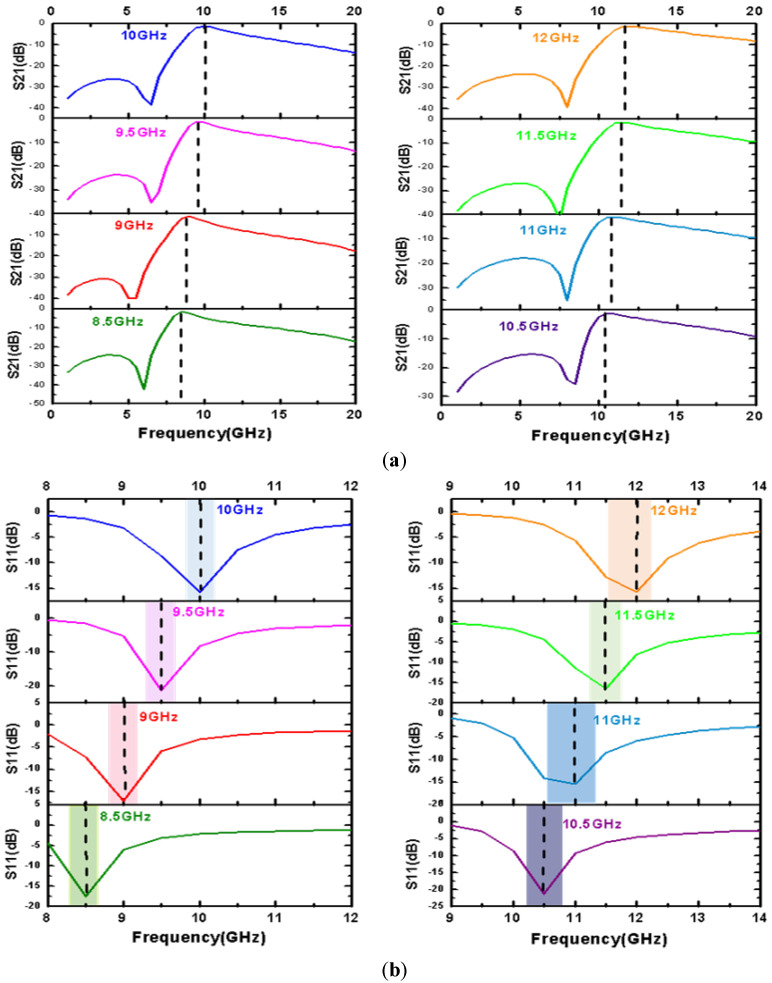
(**a**) S21 simulation results of the bandpass MEMS filter with different center frequency; (**b**) S11 simulation results of the bandpass MEMS filter for different widths of the top electrode.

**Figure 8 micromachines-14-00280-f008:**
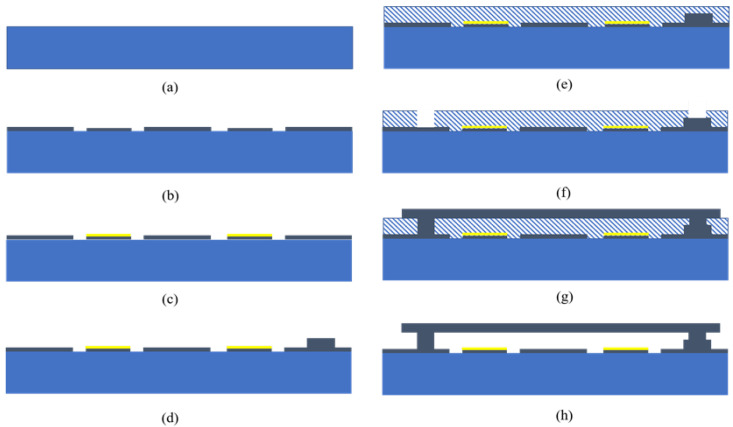
The fabrication processes of the MEMS switch: (**a**) preparation, cleaning, and quartz glass substrate cleaning; (**b**) the fabrication of CPW; (**c**) the fabrication of dielectric layer; (**d**) the fabrication of resonant ring; (**e**) sacrificial layer preparation; (**f**) making anchor through holes; (**g**) the fabrication of top electrode; (**h**) sacrifice layer release.

**Figure 9 micromachines-14-00280-f009:**
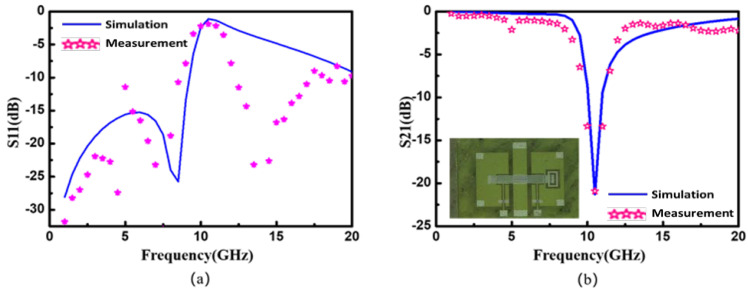
(**a**) S11 test results and simulation results; (**b**) S21 test results and simulation results.

**Table 1 micromachines-14-00280-t001:** Resistivity of some common metals.

Metal	Platinum	Aluminum	Gold	Tungsten	Copper
**Resistivity (Ω·m)**	2.22	2.83	2.40	5.48	1.75

**Table 2 micromachines-14-00280-t002:** The list of the S parameters of MEMS filter.

No.	Center Frequency (GHz)	Width of Top Electrode (μm)	Insertion Loss (dB) (S21)	Return Loss (dB) (S11)	Stopband Rejection (dB) (S21)
1	8.5	260	−1.27	−17.59	−25.83
2	9	240	−1.12	−17.23	−25.42
3	9.5	220	−1.17	−21.21	−35.11
4	10	200	−1.14	−15.68	−38.57
5	10.5	190	−1.13	−21.28	−25.03
6	11	180	−1.03	−16.54	−25.55
7	11.5	170	−1.24	−15.41	−29.12
8	12	140	−1.10	−15.69	−26.76

**Table 3 micromachines-14-00280-t003:** Filter simulation results and test results.

	Center Frequency (GHz)	Insertion Loss(dB)	Return Loss (dB)	Stopband Rejection (dB)	Drive Voltage (V)
Simulation	10.5	−1.13	−21.28	−25.03	6.48
Measurement	10.5	−1.12	−21.3	−28.3	5.80

**Table 4 micromachines-14-00280-t004:** Comparison of bandpass filters.

Ref.	Center Frequency	Insertion Loss	Stopband Rejection
2012 [28]	1.6/2.0 GHz	0.8 dB	40 dB
2015 [29]	35.32 GHz	-	20.19 dB/18.29 dB
2016 [25]	1.15 GHz	2 dB	30 dB
2018 [12]	9.5 GHz	1.2 dB	-
2019 [30]	4.5 GHz	1.7 dB	
2020 [31]	27~29 GHz	1.92 dB	23.4 dB
This work	8.5 GHz	1. 12 dB	28.3 dB
Our filter	8/8.5/9/9.5/10/10.5/11/11.5/12 GHz	≤1.86 dB	30.5 dB

## Data Availability

Not applicable.

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
