# Peer review of "Design and Fabrication of a MEMS Bandpass Filter with Different Center Frequency of 8.5–12 GHz"

_micromachines, 2023, doi:10.3390/mi14020280_

Round 1
Reviewer 1 Report (New Reviewer)
The authors present a RF MEMS bandpass filter (BPF) with variable passband center frequency in which the metalized structures are made out of aluminum. They provide the design steps, theoretical background, fabrication method and experimental results. They show mild improvement in in insertion loss of the BPF compared to Alzemi et al 2016. The concept of the design is novel and of interest for the readers of Micromachines journal. I suggest the authors bolster the introduction part with prior art and make clear where they see clear improvement and where they have regression compared to literature. A table which includes each design and their critical figure of merits (eg. center frequency, insertion loss, stop band rejection and etc) will clearly show where this design stands compared to other RF MEMS filters. The manuscript (MS) is well written however there are minor English mistakes throughout the MS that needs editing. Some other minor revisions that needs to be done before publication,
- Wherever you're using an equation you need to provide direct reference.
- In section 2.1 please provide a reference or justification on the equivalent circuit components for each MEMS component in your system.
- Figure 9 needs a legend differentiating experimental (red dots) and simulated results.
- There is a dip in the experimental S11 @ ~15GHz, how can this be explained?
Author Response

Reviewer 2 Report (New Reviewer)
In the manuscript, the authors designed and fabricated a MEMS bandpass filter with different center frequency and made some related computation on the structure. The proposed structure for tunable filter is interesting, however some problems are enumerated as follows:
1) The authors refer directly to the novelty of the work in the manuscript.
2) Authors proposed a circuit model for tunable filter which is based on Ref. [22]. It should be explained that does this model is the same as the ref. [22]? What is difference of the circuit model of the proposed structure with the Ref. [22]? Authors should calculate the values of circuit model elements.
3) What is “g1” in the equations (5) and (6). Authors should determine each of the equation parameters such as g0, g1, … .
4) What is the reason for the large difference between the fabrication and simulation results shown in Figure 9a especially in the frequency range of 10-20 GHz? Is this difference due to the weakness of the circuit model?
5) It is recommended that authors compare the theoretical and fabrication results related to the actuation voltage of the proposed structure in a table.
6) Authors provide more explanation about Eq. (2) and mention a reference for it.
7) It is better to use more up-to-date papers in the comparison table (4). Also, the quality factor of the filters should be compared. In addition, according to this table, there are not very interesting results compared to other works.
8) It is recommended to include the following articles as references in the references section to make the article more productive.
- S. M. Mirebrahimi, et al, “MEMS tunable filters based on DGS and waveguide structures: a literature review”, Analog Integrated Circuits and Signal Processing, Vol. 108, No. 1, 2021.
- S. De, et al,” Reliable, Compact, and Tunable MEMS Bandpass Filter Using Arrays of Series and Shunt Bridges for 28-GHz 5G Applications”, IEEE Transactions on Microwave Theory and Techniques, Vol.69, Issue,1,2021.
- M.M. Teymoori, et al, “A Compact Low-Loss 6-bit DMTL Phase Shifter Using a Novel Three-State Unit Cell”, Circuits, Systems and Signal Processing (CSSP), Vol.41, pp.4210-4237, 2022.
- J. Luo, et al,” Compact multiband bandpass filters based on parallel coupled split structure multimode resonators”, Electronics Letters, Volume 59, Issue 1, 2022.
- S.M. Mirebrahimi, et al, “High-quality coplanar waveguide tunable band-stop filter using defected ground structure and comb-line resonator with radio frequency microelectromechanical system varactors”, International journal of circuit theory and applications, vol.48, Issue 9,2020.
Author Response
- The authors refer directly to the novelty of the work in the manuscript.
In this paper, a bandpass filter is designed. In order to reduce the production cost, the all-aluminum process is adopted with good out-of-band suppression of insertion loss. By changing the electrode structure, the filter design with multiple center frequencies of 8.5-12GHz is realized
- Authors proposed a circuit model for tunable filter which is based on Ref. [22]. It should be explained that does this model is the same as the ref. [22]? What is difference of the circuit model of the proposed structure with the Ref. [22]? Authors should calculate the values of circuit model elements.
Both adopt double-end fixed beam, the difference is that different materials and electrode sizes are used, and many hole-like array mechanisms are designed on the electrode.
This paper is only used for circuit understanding, and the specific parameters are simulated by HFSS without quantitative analysis
- What is “g1” in the equations (5) and (6). Authors should determine each of the equation parameters such as g0, g1, … .
There is no specific numerical operation in the article,the values of the low-pass prototype filter.
- What is the reason for the large difference between the fabrication and simulation results shown in Figure 9a especially in the frequency range of 10-20 GHz? Is this difference due to the weakness of the circuit model?
It may be due to process errors and simulation conditions that the size and simulation in process manufacturing are different.
No, the equivalent circuit is only used to understand the HFSS 3D model used in the simulation.
- It is recommended that authors compare the theoretical and fabrication results related to the actuation voltage of the proposed structure in a table.
Has been modified in the article
- Authors provide more explanation about Eq. (2) and mention a reference for it.
Where l represents the length of the upper electrode, w represents the width of the upper electrode, t represents the thickness of the upper electrode, and E represents the Young's modulus of the upper electrode material
7) It is better to use more up-to-date papers in the comparison table (4). Also, the quality factor of the filters should be compared. In addition, according to this table, there are not very interesting results compared to other works.、
The quality factor is not the key factor in this paper, and the quality factor is rarely mentioned in the literature mentioned
8) It is recommended to include the following articles as references in the references section to make the article more productive.
Has added in article
- S. M. Mirebrahimi, et al, “MEMS tunable filters based on DGS and waveguide structures: a literature review”, Analog Integrated Circuits and Signal Processing, Vol. 108, No. 1, 2021.
- S. De, et al,” Reliable, Compact, and Tunable MEMS Bandpass Filter Using Arrays of Series and Shunt Bridges for 28-GHz 5G Applications”, IEEE Transactions on Microwave Theory and Techniques, Vol.69, Issue,1,2021.
- M.M. Teymoori, et al, “A Compact Low-Loss 6-bit DMTL Phase Shifter Using a Novel Three-State Unit Cell”, Circuits, Systems and Signal Processing (CSSP), Vol.41, pp.4210-4237, 2022.
- J. Luo, et al,” Compact multiband bandpass filters based on parallel coupled split structure multimode resonators”, Electronics Letters, Volume 59, Issue 1, 2022.
- S.M. Mirebrahimi, et al, “High-quality coplanar waveguide tunable band-stop filter using defected ground structure and comb-line resonator with radio frequency microelectromechanical system varactors”, International journal of circuit theory and applications, vol.48, Issue 9,2020.

Reviewer 3 Report (New Reviewer)
This paper is written well and the work is interesting. The work is technically sound and worthy for Publication.
Author Response
Modifications are marked in red

This manuscript is a resubmission of an earlier submission. The following is a list of the peer review reports and author responses from that submission.
Round 1
Reviewer 1 Report
This paper presents design, simulation, and fabrication of a MEMS bandpass filter with different center frequencies ranging from 8.5 – 12 GHz. The micromachine characteristics of the filter are optimized to change the center frequency. A filter worked at 10.5 GHz is fabricated with aluminum top electrode which exhibits low insertion loss as well as high stopband rejection.
The experimental setup is clear; abstract and conclusion should be revised with concise explaination. The following comments should also be addressed for the publication of the manuscript:
- In the Introduction, In the Introduction, the literature on integrated filter devices should be enlarged, highlighting the flat-bandwidth (New microwave photonic filter based on a ring resonator including a photonic crystal structure. In 2017 19th International Conference on Transparent Optical Networks (ICTON) (pp. 1-4). IEEE, 2017.) or high ER (e.g., Silicon graphene reconfigurable CROWS and SCISSORS. IEEE Photonics Journal, 7(2), 1-9., 2015; High performance and tunable optical pump-rejection filter for quantum photonic systems. Optics & Laser Technology, 139, 106978, 2021) or small linewidth (e.g, Rigorous model for the design of ultra-high Q-factor resonant cavities. In 2016 18th International Conference on Transparent Optical Networks (ICTON) (pp. 1-4). IEEE., 2016; Integrated waveguide coupled Si 3 N 4 resonators in the ultrahigh-Q regime. Optica, 1(3), 153-157., 2014; Ultralow 0.034 dB/m loss wafer-scale integrated photonics realizing 720 million Q and 380 μW threshold Brillouin lasing. Optics letters, 47(7), 1855-1858, 2022).
- Abstract should be revised with clear and precise descriptions of all key findings.
- Conclusion should be revised with clear and precise descriptions of all key findings.
- In line 11, grammatical error of “the”
- In line 13, name the other electrical parameters
- In line 14, grammatical error, use correct form “working”
- In line 15, change “exhibits” with another synonym
- In line 16, break the sentence, precisely describe simulation results and what are the performance parameters, mention them.
- In line 18, which electrical properties are improved mentioned them.
- In line 25, “especially have distinguished cognitive radios from radios,” statement is not cleared.
- Reference 12 need revision to clearly state the description.
- In line 29, name the electronic devices and add some references.
- In line 34, reference 15, mention numerical value of insertion loss.
- In line 38, grammatical error of word “geometric”, change it with geometrical.
- In line 40, the word “purpose” is not appropriate, change it.
- In line 41, difficulty of fabrication, need grammar check.
- In line 42, explain how aluminum structure reduce the production cost. Justify it.
- In line 44, grammar error of using “An”, no need of it.
- In line 45, repetitive use of “an”, replace it.
- In line 46, 47, mention numerical value with specific amount term.
- In line 52, how “resonant ring is the most important structure in the RF MEMS switch,” explain it.
- In line 56, no need to write “in addition”.
- In line 61, 62, mention maximum value of voltage and displacement.
- In line 70, grammar check of “,” after voltage.
- In line 73, clearly mention which area of electrode is discussed.
- Labels of Figure 2 are not clear, change it for clear vision.
- In line 82, grammatical error of using “and,” change it.
- In line 83, s should be small in word “Straight”.
- In line 89, change the word “specific”, repetitive here.
- In line 90, Stopband Rejection has grammar error of first letter.
- In line 97, no need of “,” after loss.
- In line 111, 112, use abbreviation CPW.
- In line 118, follow same pattern for Figure description as previously.
- In line 119, grammar error of “,” after converter.
- In line 122, follow same pattern for Figure description as previously.
- In line 124, follow same pattern for Figure description as previously, check for grammar of word “admittance converter” and use “:” at the end of line 124.
- In line 145, add table for material properties comparison of aluminum with other materials to justify.
- In line 148, grammar check for word “critical”.
- In line 151, follow same pattern for Figure description as previously.
- In line 154, grammar check for word “geometric”.
- In line 156, remove this sentence.
- In line 157, what is S parameters, add details.
- In line 165, mention the name of process.
- In line 167, grammatical error of using “.”, remove it.
- In line 168, follow same pattern for Figure description as previously.
- In line 177, change word “then” with “finally”.
- In line 180, add space before “Si3N4”.
- In line 186, change word “then” with “finally”.
- In line 192, 193, too short sentences, loosing continuity, rephrase it.
- In line 199, 200, should be a single sentence and end at 4000 r/min.
- In line 203, change word “then” with “finally”.
- In line 212, change the word “connect”, seems inappropriate.
- In line 214, mention the name of calibration tool.
- In line 215, remove the term “of the filter,” and also change “thereby” with another synonym.
- In line 220, rephrase the sentence and remove “as shown” and add “.” at the end.
- In line 224, add space after “Table 3”.
- In Table 3, add some most recent papers for comparison.
- In line 233, rewrite sentence by adding design, simulation terms which have been performed in paper.
- In line 236, use “and” instead of “,” and replace “S” with “s” in stopband.
- In line 237, add “is” after “GHz” and add “which” instead of “,” after “electrode.”
- In line 238, break the sentence after 28.3 dB and start a new sentence with brief description of which electrical performance parameters are evaluated.
- Also mention how cost parameter is reduced with this design.
Author Response
The comments of reviewers have been revised in the original text, and the revised parts have been marked in red.

Reviewer 2 Report
The submitted manuscript reports the design, fabrication and characterization work on a MEMS bandpass filter device at a designed working frequency in the range of 8.5-12 GHz. Although the topic of this paper falls into the journal’s scope, the manuscript seems to be unfinished at submission. For example, there are two Figure 7, plus another figure with no caption. There are a lot of format errors (e.g. numbers in some chemical formulae and symbols are not subscripted, no space between values and units). The references of the equations used in this manuscript are not provided. Labels in Fig. 1 need to be double-checked. Language needs to be (e.g. ‘an actuation electrode’ appeared twice in one sentence, page 1 / line 44-45). The referee does not recommend this manuscript be published in its current form.
Considering that the storyline of the manuscript is complete, the referee suggests the authors resubmit their manuscript after the authors revise it thoroughly.
Author Response
The comments of reviewers have been revised in the original text, and the revised parts have been marked in red

Round 2
Reviewer 1 Report
The Authors have modified the manuscript according to the Reviewer comments. I suggest to format the references as requested by Micromachines journal.
Author Response
1)The format of citations has been revised.
2)The subscripts of chemical symbols and special characters have been modified.
3)Already made all notation formatting consistent.
4)The name of Figure 1 has been modified.
5)Modifications have been made in page 1 / line 44-45

Reviewer 2 Report
The authors ignored most of the comments from the referee (references, formats, language) in their revised version.
Author Response

(The authors gave the same response as above.)
